# An Adaptable DTS-based Parametric Method to Probe Near-surface Vertical Temperature Profiles at Millimeter Resolution

Constantijn G.B. ter Horst<sup>1</sup>, Gijsbert A. Vis<sup>2,3</sup>, Judith Jongen-Boekee<sup>2</sup>, Marie-Claire ten Veldhuis<sup>2</sup>, Rolf W. Hut<sup>2</sup>, and Bas J.H. van de Wiel<sup>1</sup>

**Correspondence:** Bas J.H. van de Wiel (b.j.h.vandewiel@tudelft.nl)

Abstract. We present a novel, fine-resolution temperature profiling method based on Distributed Temperature Sensing (DTS) that is adaptable, reproducible, and fully FAIR. Accurate probing of near-surface temperature gradients requires sub-centimeter resolution, particularly in environments with short vegetation such as grass, where strong insulating properties promote steep gradients. Conventional DTS systems provide spatial resolutions of approximately 25 cm along fiber optic cables that can span several kilometers. By compacting such cables into a helical coil supported by a laser-cut frame, the Fine Resolution Adaptable Distributed Temperature Sensing (FRADTS) method attains vertical resolution and accuracy at the millimeter scale. The frame design is generated by a parametric script that outputs laser-cutting files, enabling users to assemble coil structures from sheet material with identical or easily adapted geometries. We demonstrate the method in both laboratory tests and a field campaign at the CESAR atmospheric observatory in Cabauw, the Netherlands, where a prototype coil captured high-quality vertical temperature profiles within the lowest meter above the soil, including a 10 cm grass layer. A resolution of 1.3 mm was attained and verified, and the influence of environmental factors such as solar radiation and precipitation on measurement biases was mapped and quantified. Despite minor artifacts, the method proved robust and effective, providing high-quality profiles under a wide range of weather conditions. As the method is modular and parametric, it can easily be applied in other research, potentially extending its application to other fields.

#### 5 1 Introduction

An accurate description of near-surface temperature profiles is important for atmospheric modeling. One might want to predict near-surface phenomena such as ground fog (Gultepe et al., 2007) and crop damage due to frost (Snyder and de Melo-Abreu, 2005), to proactively enact mitigation strategies. Surface temperatures are often not directly measured, but typically they are extrapolated from models, usually under the assumption that near-surface profiles attain a (quasi) logarithmic shape (Foken, 2006; Moene and van Dam, 2014). These logarithmic relations stem from the fact that turbulent transport near rigid surfaces is limited due to the fact that the characteristic eddy size decreases closer to the surface. This is referred to as the 'law of the wall' (Bradshaw and Huang, 1995). It results in increased vertical temperature gradients near the surface. However, it has been well established that, within the roughness sublayer (RSL), which is the atmospheric layer where roughness elements have a

<sup>&</sup>lt;sup>1</sup>Delft University of Technology, Department of Geoscience and Remote Sensing, Delft, The Netherlands

<sup>&</sup>lt;sup>2</sup>Delft University of Technology, Department of Water Management, Delft, The Netherlands

<sup>&</sup>lt;sup>3</sup>GFZ German Research Centre for Geosciences, Section Hydrology, Potsdam, Germany

significant influence on turbulent transport, temperature profiles start to depart from the logarithmic behavior that is typically found further aloft. This discrepancy may have many physical causes: longwave radiative cooling, local latent heat effects (fog formation), but it may also be due to aerodynamic effects around individual grass tussocks. In any case, extrapolation of the log-law towards the surface is non-physical as it would imply infinite vertical gradients at the surface. In spite of this, extrapolation of this model down to the surface is still common in atmospheric modeling (Zhang et al., 2022; Meier et al., 2022). This has led to the development of alternative models for near-surface temperature profiles (Boekee et al., 2024).

To test, validate and improve such models, fine-scale temperature observations near the surface are crucial. However, vegetative canopies add an additional layer of complexity to near-surface temperature profiles, as they also act as an insulating layer, separating the soil from the atmosphere. Particularly in short canopies, such as grass, the insulative properties result in exceptionally steep temperature gradients near the surface, requiring extremely high vertical probing resolutions on the order of millimeters (Izett et al., 2019). Moreover, these steep temperature gradients occur at many other atmospheric interfaces as well, such as in snow (Zeller et al., 2021). Resolution and accuracy on this scale cannot be achieved with conventional point measurements using temperature probes. As a consequence, profiling the temperature near the surface at a sufficient resolution to capture these gradients becomes a challenge.

30

Advances in temperature surveying technology have resulted in the development of the method of Distributed Temperature Sensing (DTS) to probe temperatures within fiber optic cables, on the basis of Raman scattering. The DTS method provides spatial resolutions along the fiber of up to 12.5 cm, which results from the sampling frequency of time-of-flight measurements. The DTS method provides temperature data resolutions on the order of seconds. This technology is capable of monitoring temperature profiles over cables spanning several kilometers in length, making it suitable for applications requiring high-resolution thermal mapping (Ukil et al., 2012). Fiber optic temperature can be used as a proxy for air temperature, thanks to the low thermal mass of the cable ensuring fast equilibration with the surrounding air (de Jong et al., 2015). Furthermore, fiber optic cables can be deployed in compact configurations, resulting in higher effective spatial resolutions than the DTS inherently offers.

Temperature profiles from the soil to the top of tall canopies such as forests, have already been successfully studied using DTS-based measurement setups (Schilperoort et al., 2020; Peltola et al., 2020). For these studies, probing temperature profiles at a spatial resolution on the order of 10 cm was sufficient, as the tall canopies did not require densely spaced data. Additionally, some studies, such as Boekee et al. (2024), have attempted to resolve near-surface temperature profiles in short (grass) canopies. This campaign required much more densely spaced data as a consequence of the canopy being merely centimeters in height. Therefore, adapted, compactly positioned fiber optic configurations were implemented to attain sufficiently high resolution vertical temperature profiles. Fiber optic cable was attached to a net to form a two-dimensional harp-like structure. Additionally, novel configurations have been developed where helically wound coils further increase vertical fiber density (Hilgersom et al., 2016; Sigmund et al., 2017; Izett et al., 2019; Zeller et al., 2021). These campaigns implemented rigid, vertical frames, around which the fiber optic cable is wound. As a result of these enhanced geometries, resolutions and accuracies up to the centimeter scale were achieved.

These campaigns were successful in achieving their respective specific goals, such as detecting shallow fog and qualitatively comparing RSL models. However, the campaigns implemented configurations which introduce uncertainties in the way they are assembled and deployed, making the data difficult to reproduce accurately.

The vertical probing resolution is a critical limitation in quantifying temperature gradients. Previous campaigns have achieved resolutions of several centimeters, but capturing near-surface gradients within the upper meter requires sub-centimeter resolution and accuracy. Beyond the inherent spatial resolution of DTS devices, this limitation is largely geometric in nature and can be divided into two aspects. The first is that of **vertical fiber density**: increasing the length of fiber per unit vertical distance enhances resolution. This issue is addressed by the enhanced fiber geometries, such as the aforementioned coil geometry. The second aspect is that of **positional accuracy**: the temperature measurement at any point depends directly on the precise location of the fiber, which requires the cable to be fixed securely in space. Even with increased vertical density, uncertainties in fiber positioning remain a limiting factor. Thus, further increasing fiber winding density in isolation is insufficient; both vertical density and positional accuracy must simultaneously be addressed to achieve meaningful improvements in vertical resolution.

Previous campaigns also encountered issues due to the frame holding the fiber optic cable that introduces a significant bias into the data. This bias in temperature was mainly due to the absorption of solar radiation by the frame, in turn leading to heating of the fiber optic cable (Sigmund et al., 2017). This resulted in artifacts in the data, in addition to an overall bias towards warmer temperatures. Also, temperature profiles may be influenced by evaporative cooling of the wet frame after rain events. This introduces similar artifacts and biases. Furthermore, the frame undergoes radiative cooling at night, which introduces cooling artifacts. Finally, in spite of its open structure, the presence of the frame affects the wind flow through the coil, which also has an effect on the local cable and air temperatures.

70

There is a need for a more robust and universal method of probing temperature profiles with DTS, in such a way that data is representative of local air temperature, reproducible, and useful in a wider range of applications. In this paper, we present a new approach which addresses the aforementioned issues in such a way that accurate and repeatable DTS-based measurement devices can be deployed in the field, with a probing resolution and accuracy up to the mm scale. We have designed and tested a DTS-based temperature profiling method which can be universally implemented across different fields in a way that is accurate, consistent, low-cost, modular, adaptable and easily manufactured. The design process was conducted in accordance with the FAIR guiding principles for scientific data sharing, wherever they were applicable to hardware development (Wilkinson et al., 2016). Here, the aim is not only to design a single setup for one specific application, but rather develop a method that can be adapted to other applications. Thus, the proposed method makes high-resolution data for temperature research much more accessible and reproducible. This potentially extends such high resolution observations to applications beyond the grass region towards other media, such as snow (Zeller et al., 2021), ice (De Bruijn et al., 2014) or water surfaces (Paaijmans et al., 2008).

In this paper, firstly, a design is formulated on the basis of a set of criteria in section 2. The final design is directly described, along with key design choice justifications. Subsequently, a prototype is built, tested and evaluated, both in lab and field conditions described in section 3. The resulting observations are compared to benchmark observations and discussed in section 3. Finally, design optimization and practical suggestions are given in section 4.

# 2 Design

The effective development of a robust, high-resolution, DTS-based temperature profiling method requires an elaborate design process. In this process, iterative steps are taken to reach a final product. The measurement method will hereafter be referred to as 'the design', though it refers to the development of a method rather than a singular device. As the method will be fundamentally based around DTS, a structure which accurately positions the fiber optic cable will be needed. This structure is referred to as 'the frame'. This section outlines the results of the design process and describes the final method. Summarizing the introduction: we will be designing a solution that allows us to probe near-surface vertical temperature with fiber optic DTS at an unprecedented resolution in the vertical. The design will be modular and adjustable to facilitate others replicating and adapting it to their own research.

#### 2.1 Criteria



In this section a set of design criteria is presented. A final design which satisfies all of these criteria is considered to be successful as it fits the overall goals described in the previous paragraph. Criteria can be subdivided into two categories: threshold criteria and optimization criteria. Threshold criteria involve a binary evaluation, where a design either passes or fails based on predefined thresholds. In contrast, optimization criteria focus on optimizing multiple design parameters to achieve the best possible performance relative to the specified criterion. Optimizing performance in one criterion often comes at the expense of performance in others, resulting in a trade-off. A list of relevant criteria is given in Table 1 and Table 2.

**Table 1.** Threshold criteria for DTS frame design

| Criterion              | Description                                                                                                                                                                                                                                                                                                                                                                                                                                                                                                                                                                                                               |  |
|------------------------|---------------------------------------------------------------------------------------------------------------------------------------------------------------------------------------------------------------------------------------------------------------------------------------------------------------------------------------------------------------------------------------------------------------------------------------------------------------------------------------------------------------------------------------------------------------------------------------------------------------------------|--|
| Temperature resolution | The design must allow for the measurement of 1-dimensional vertical temperature profiles at a vertical resolution and accuracy required for the application at hand. For the application presented here, steep temperature gradients in a grass layer, we set the minimum requirements at 5 and 2 mm, respectively. Fiber temperature accuracy and temporal resolution are inherent to the DTS device itself and therefore do not fall within the scope of this criterion. However, depending on implementation, they can reasonably be expected to reach up to 0.1 °C and 5 s respectively (Van de Giesen et al., 2012). |  |
| Weather resistance     | The design must be able to withstand extreme weather conditions for at least 3 months in the field:  - Ambient temperatures between -20 °C and 40 °C  - Wind gusts of up to 20 m/s  - Daily solar radiation of up to 1200 W/m <sup>2</sup> - Full water submersion in case of heavy rainfall                                                                                                                                                                                                                                                                                                                              |  |
| Reproducibility        | The design must be reproducible.                                                                                                                                                                                                                                                                                                                                                                                                                                                                                                                                                                                          |  |
| Ease of construction   | The design must be constructible without any prior fabrication experience.                                                                                                                                                                                                                                                                                                                                                                                                                                                                                                                                                |  |
| Accessibility          | The design must be fabricated using tools and materials which are commonly accessible at research facilities or university campuses where DTS devices are also available.                                                                                                                                                                                                                                                                                                                                                                                                                                                 |  |

Table 2. Optimization criteria for DTS frame design

| Criterion                       | Description                                                                                                                                                                                                                                                                                                                                                                                                                                                                                                                                                                                                                                     |  |
|---------------------------------|-------------------------------------------------------------------------------------------------------------------------------------------------------------------------------------------------------------------------------------------------------------------------------------------------------------------------------------------------------------------------------------------------------------------------------------------------------------------------------------------------------------------------------------------------------------------------------------------------------------------------------------------------|--|
| Representativeness              | The temperature measurements must represent the local temperature of the medium in which the design is placed, to the highest degree possible. Therefore, the thermal mass and conductivity of the frame need to be minimized, to lower the equilibration time of the intrsument to the medium temperature and to ensure that the frame is not moving heat across the temperature gradient that is being measured. Furthermore, the effects of external factors such as solar radiation absorption and evaporative cooling due to precipitation should be minimized. For example, frame albedo must be maximized to mitigate radiative heating. |  |
| Adaptability                    | Physical parameters of the design, such as height, width and resolution need to be easily adaptable. All relevant parameters should be adjustable to make the design more adaptable and universal.                                                                                                                                                                                                                                                                                                                                                                                                                                              |  |
| Minimal environmental influence | The frame should obstruct as little solar radiation as possible and be as permeable to wind as possible to reduce influence on local medium-flow.                                                                                                                                                                                                                                                                                                                                                                                                                                                                                               |  |
| Cost efficiency                 | Design cost must be minimized.                                                                                                                                                                                                                                                                                                                                                                                                                                                                                                                                                                                                                  |  |
| Fabrication efficiency          | Fabrication time must be minimized.                                                                                                                                                                                                                                                                                                                                                                                                                                                                                                                                                                                                             |  |

# 2.2 Design outline








Several design iterations were made and evaluated on the basis of the criteria outlined in subsection 2.1. The iterations led to the formulation of a final design, which will be directly presented here as Fine Resolution Adaptable Distributed Temperature Sensing (FRADTS). The proposed method is based on the technique of laser cutting, which has become an increasingly accessible manufacturing tool. Laser cutters are available at most modern workshops and makerspaces. Furthermore, many commercial laser cutting services exist. It enables users to cut a thin sheet of material along a specified path at a high precision of around 0.1 mm. The cut path is defined in files which are generated using CAD software. This, in turn, allows for the automatic generation of these files, so that cut paths are calculated from a set of input parameters. This is known as *parametric design* (Casini, 2022) and forms the fundamental basis of FRADTS.

The proposed method is a tool in the form of a software script which generates laser cutting files from a broad range of input parameters, which can be specified by the user. With the resulting files, individual parts can be cut out of a flat sheet of material. The parts can be assembled into a 3 dimensional frame which holds the DTS fiber firmly in place, with high positional accuracy. Input parameters such as physical dimension and resolution can be adjusted, resulting in full adaptability for different applications. Furthermore, FRADTS requires little to no manufacturing experience or skill, as the parts are easily put together. At the same time, a high degree of reproducibility is attained, as the exact same parts result from a given set of input parameters.

# Frame geometry

The proposed geometry is a lightweight, rigid frame composed of a rib structure that holds a coiled fiber optic cable in place, as depicted in Figure 1. A number of vertical ribs are held together in a circular configuration using several support rings into which these ribs are slotted. The ribs have an array of notches on their leading edges which hold the fiber optic cable in place under tension, as demonstrated in Figure 2. These notch arrays are shifted vertically with every consecutive rib, forming an upward spiral as the fiber is wound onto the support ribs. This results in consistent fiber distance, yielding high positional accuracy. The fiber is clamped at the entrance and exit points of the coil using luster screw terminals to keep it tensioned. The coil can optionally be embedded into soft surfaces, such as soil, using long stakes at the bottom of the frame. For hard surfaces, such as concrete, the stakes can also be omitted, placing the coil directly onto the surface. For additional rigidity, the top of the coil can be tethered to the ground using thin nylon wire.

### **Material selection**

The laser cutting method comes with a range of compatible sheet materials, each with its own advantages and drawbacks. As part of the design process, an optimal material for the frame is to be selected for deployment in the field. Several factors play a role in material selection, which are stipulated in the design criteria.

Most importantly, the frame should have minimal thermal influence on the cable that it holds. To reduce heat exchange between the frame and fiber, two parameters have to be minimized. The first parameter is the volumetric heat capacity, given by the product of the density and specific heat of the material:  $\rho c_p$  [J/m³/K]. This parameter determines the amount of heat that can be stored within a given volume. Reducing the amount of heat stored within the frame diminishes the potential heat

**Figure 1.** Top view of the final coil geometry. Here the structure of the frame is clearly visible. Vertical support ribs are slotted into a series of rings, which form a rigid frame using a minimal amount of material. The DTS fiber optic cable is wound around the frame and spirals helically upward to its top.

transfer to the fiber optic cable, decreasing equilibration time and thus improving response speed. Secondly, the thermal conductivity of the frame:  $\lambda$  [W/m/K] is to be minimized. This parameter determines the rate at which heat, stored within the frame, is transported into the fiber and vice versa. To minimize both parameters, an Ashby plot (Ashby, 2025) was used to determine the optimal material, keeping in mind discrete threshold criteria, such as strength, weather resilience and machinability (GrantaEduPack, 2025). Figure 3 depicts a range of potential frame materials. Unsuitable materials are depicted in gray, as they did not meet the threshold. Materials with ideal thermal properties tend towards the bottom left of the figure. The figure indicates that polymers have advantageous thermal properties. From this material group, polymethyl methacrylate (PMMA), a robust polymer, has additional advantageous properties. PMMA has a high UV resilience (Rahman et al., 2020) and is commonly available in thin sheets which can be processed in a laser cutter. Furthermore, the sheets are available in high-gloss, white variants which have a high reflectivity for solar radiation, maximizing albedo. The material is waterproof and is highly stable, even in case of constant submersion (Fernández et al., 2023). Finally, the material is robust within the temperature

**Figure 2.** Close up of the coil. The fiber optic cable is kept firmly in place due to the notches on the ribs. Fiber position and spacing is accurately defined as can be seen in the image. The distance between the fibers is very consistent, meaning that its position is precisely defined.

limits specified in the criteria (Abdel-Wahab et al., 2017). This makes it an especially suitable material for this application. It is therefore recommended to use PMMA as the frame material for deployment in the field. However, due to the modular nature of the method, PMMA can easily be substituted for other materials if alternative characteristics are required.

# Parametric design



We implemented parametric design by writing a novel script which generates a cutout template for the coil, which can be traced by a laser cutter. This program takes a set of input parameters to generate this template. The script is developed in Grasshopper, which is an extension of the Rhinoceros 3D CAD software, specifically aimed at parametric design (Robert McNeel and Associates). In this software, many parameters can be adapted to change the physical properties of the frame. A selection of the most relevant parameters are listed in Table 3. Example values are included in the right column. These parameter values were used in our experimental validation of FRADTS. The most important property of the setup, the vertical resolution, is calculated as follows:

$$dz = \frac{H_{coil}\Delta l_{DTS}}{2\pi r_{coil}N_{windings}} \tag{1}$$

Here,  $r_{coil}$  is the radius of the coil [m],  $N_{windings}$  is the number of windings around the coil,  $H_{coil}$  is the height of the coil [m] and  $\Delta l_{DTS}$  is the spatial resolution along the fiber of the DTS device [m].

The implementation of parametric design makes the method reproducible and adaptable, as was part of the initial criteria. The only requirement for identical reproduction of a given setup is the unique set of input parameters. Furthermore, the parameters can be quickly changed for rapid prototyping of different setups. As a result, FRADTS is applicable to a broad range of research applications.

**Figure 3.** An Ashby plot depicting potential materials to be used for the frame. Materials that do not meet threshold criteria for strength, weather resilience and machinability are depicted in gray. Materials to the bottom left have the best thermal performance and minimize heat transfer between the frame and the fiber. Polymers, depicted in blue, are the material group closest to this region.

# 2.3 Coil fabrication and setup


Fabricating and installing the coil is a relatively simple process, which requires little practical experience. Assembly requires running the script, laser cutting the paths, slotting the resulting parts together and winding the fiber optic cable around the frame. The full fabrication process is described in detail in the project repository (linked under 'code availability', section 5). A short summary of the process is given below.

Firstly, the desired physical parameters of the coil are configured in the grasshopper script, which generates a laser file. The desired sheet material (PMMA is highly recommended for field deployment) is put into the laser cutter, which traces the lines in the file. This results in a collection of flat puzzle-like pieces, which form the ribs and rings of the frame. The pieces are slotted together and are glued for extra rigidity (Figure 4). Then the fiber optic cable is wound around the frame, retaining sufficient cable lengths at the ends for calibration. The cable is secured by luster terminals in the ribs . The coil is now finished and ready to be deployed in an experiment.

The coil can be installed in many different environments. For the experimental validation of FRADTS, deployment on a grassy field was chosen (Figure 5). To install the coil in the field, holes were pre-made in the soil using a stencil and a drilling

| Coil parameters                    | Value      |
|------------------------------------|------------|
| Coil height                        | 1000 mm    |
| Number of windings                 | 200        |
| Coil radius                        | 159.155 mm |
| Number of support ribs             | 10         |
| Number of support rings            | 6          |
| Support rib width                  | 13 mm      |
| Support ring width                 | 13 mm      |
| Fiber optic radius                 | 0.750 mm   |
| Sheet material thickness           | 3 mm       |
| Bottom stake length (optional)     | 400 mm     |
| Number of rib sections (optional)  | 2          |
| Number of ring sections (optional) | 2          |

**Table 3.** A subset of the most relevant input parameters to the file generation script. The resolution of the coil is determined by the number of windings and the coil height. In this example, a 1000 mm coil height with 200 windings results in 5 mm vertical spacing between the windings and 1.25 mm measurement resolution along the vertical. Note that the minimum coil radius is constrained by the minimum bend radius allowed by the type of fiber that is used. Support rib and support ring widths determine how much material is used for the coil. They can be optimized, such that the structure is sufficiently strong, while minimizing the volume of the frame. Generation of the bottom stakes can be turned on by specifying a length. They will be omitted if set to 0. Finally, the support ribs and support rings can be optionally subdivided into multiple sections. This allows for efficient material use, as well as coil dimensions which exceed the limits of the laser cutter work area.

stake. The PMMA bottom stakes are inserted into the pre-drilled holes. Subsequently, the coil is placed on top of the stakes.

Finally, the top of the coil is tethered to the soil in four directions using thin nylon wire for additional rigidity. From this point, the coil is configured with a DTS device as with any other DTS-based setup. The results from this field campaign will be further discussed in subsection 3.2.

# 3 Experimental validation and discussion

To assess the performance of FRADTS, the coil was tested in two settings. It was first evaluated in several lab experiments. In these tests, several properties were examined, such as vertical accuracy and sensitivity to precipitation and radiation. Subsequently, the coil was deployed in a field campaign to fully validate the outdoor effectiveness of the design. In both scenarios, a Silixa Ultima M DTS machine was used, which has a spatial resolution along the fiber of 25 cm. It is configured to capture a temperature profile every 5 seconds. A double-ended configuration is used to allow for calibration using differential attenuation along the fiber (Van de Giesen et al., 2012). Loosely buffered, 1.6 mm diameter, fiber optic cable with Kevlar reinforcement

**Figure 4.** The parts of the frame being assembled. Ribs are slotted into the array of support rings one by one to construct the frame. Note that additional photo material on the manufacturing and lab-tests is available in ter Horst (2024)

**Figure 5.** The coil installation process in the field. Firstly, an MDF template is used to pre-drill stake holes using a metal stake (a). Secondly, the individual stakes are pushed into the soil (b). The coil is then slotted into the tops of these stakes and fastened using the appropriate glue (c). The top of the coil is tethered to the soil by spanning 0.5 mm nylon fishing wire in four directions. Panel (d) depicts the coil installed in the field.

and white cladding was used for both settings. Manufacturer supplied Pt-100 temperature probes were used as reference, in the lab experiments to infer air and water temperatures, and in the field campaign as reference for two calibration baths.

### 3.1 Lab conditions






To assess whether the initial threshold criteria have been met, several experiments were conducted in the lab under controlled conditions.

The central criterion for the development of FRADTS relates to the vertical resolution and vertical accuracy of the design. The first parameter is determined by a design choice, as the coil height and number of windings determine the number of windings per unit length. The choice of these parameters in the coil validation prototype is stated in Table 3. Combining the amount of windings per meter with the inherent DTS resolution of 25 cm yields a vertical probing resolution of 1.25 mm for this coil. The second parameter -the vertical accuracy- is determined by multiple factors. This includes the positional accuracy of the fiber and the thermal conductivity of the frame and fiber. In order to determine the vertical accuracy, a lab experiment was conducted. The coil was positioned in an empty vertical container. After some time, cold ice water was siphoned into the bottom of the container. As a result, the water level gradually rose along the height of the coil at a constant rate. The difference in temperature between the water and ambient air resulted in a strong temperature gradient at the water-air interface. The position of this large gradient can be found by calculating the numerical gradient and tracking the spatial position of its maximum. The deviation of these positions with respect to the true water level position provides a measure for the positional accuracy of the method. The mean absolute error is taken to be the positional accuracy of the configuration. The physical setup and the shifting gradient are depicted in Figure 6. From this experiment, a positional accuracy of 1.3 mm is found, which is almost exactly equal to the vertical resolution of the coil. This confirms that probing close to a millimeter resolution is meaningful, as we achieve an accuracy on that same order.

#### 215 3.2 Field conditions

#### 3.2.1 Field setup

The lab tests confirmed that the coil passed the essential threshold criteria. To fully validate the outdoor performance of FRADTS, the coil was deployed in a field campaign at the Cabauw Experimental Site for Atmospheric Research (CESAR) observation site (A. Apituley, 2024), which is operated and maintained by the Royal Netherlands Meteorological Institute (KNMI). The site is located in Cabauw, in a polder near Utrecht in the Netherlands (51.971° N, 4.927° E), which is well suited for observations as it is located in a relatively homogeneous, mostly agricultural area. Grass at the Cabauw site is mowed and maintained at approximately 0.1 m height. It is noted here that grass height may vary somewhat and as such the 0.1 m is more indicated than absolute. As many atmospheric measurements are done here, the DTS temperature profiles are supplemented with auxiliary data which can be used for validation. Air temperature data (°C) is available at 1.5 and 0.1 m above the surface. These temperatures are recorded by shielded EPLUSE-Pt1000 and KNMI-Pt500 elements, respectively. They are both housed in open huts with only natural ventilation. The accuracy and resolution is 0.1 °C for both sensors. Data on precipitation amount (mm) is gathered by a calibrated KNMI rain gauge. The setup is wind-shielded and measures precipitation at an accuracy and resolution of 0.2 and 0.1 mm, respectively. Wind speed data at 1 meter above the surface is available in the surface fluxes dataset. This quantity is measured by a Gill R3 sonic anemometer in m/s, at a resolution and accuracy of 0.001 m/s and 

**Figure 6.** Experiment to determine the vertical accuracy of the coil. The physical setup is depicted in panel (a). The coil is placed into a leveled vertical container which is slowly filled with ice water. This results in a strong temperature gradient which moves over time. A graphical representation of the moving gradient position can be seen in the plots in panel (b). Finally, the coordinate of the maximum gradient is tracked and plotted over time in panel (c). Note that the coil was positioned upside down in the container. Hence, the water interface starts at z=1, relative to the base of the coil. In the experiment, some deviation in the data is observed with respect to the actual water level coordinate. From this deviation, a positional accuracy of the coil is derived (see text).

RMS, respectively (Gill Instruments, 2014). Finally, surface net radiation (W/m²) is measured by a Schulze net radiometer type LXG055, which measures at an estimated accuracy of 5%, as specified in metadata. (Bosveld, 2020)

The coil was installed in a grass-covered field to survey near-surface temperature profiles over the course of several months. The observation period ranges from 26-03-2024 to 22-07-2024. There exist significant gaps in the data because of DTS malfunction. As a result, a total of 36 days were recorded, spread over this period. This amount of data is still considered to be sufficient for validation as many different weather conditions were surveyed during the campaign. These include wind gusts


up to 10.8 m/s, rainfall up to 45 mm/h, air temperatures from 2.4°C to 21.7 °C and net surface radiation up to 640 W/m<sup>2</sup>. Many of these weather conditions do not exceed the ranges specified in the criteria, as these parameters cannot be controlled. However, given that the structural integrity of the frame has not been compromised by weather conditions during the full length of the campaign, FRADTS is shown to meet a wider range of the weather-related criteria.

The coil was linked to other DTS setups, in series, by splicing the ends of fiber optic cable together. The coil was included in the middle of a total of three setups. The addition of other setups required calibration after each splice. The coil consists of a section of approximately 200 m of cable and was located roughly in the middle of the 650 m total cable length. For calibration, we follow the best practices outlined in Van de Giesen et al. (2012): After the installation process, as depicted in Figure 5, 10 m sections of both ends of the fiber optic cable were run through a pair of water-filled calibration baths. One of these baths was kept at ambient temperature, while the other bath was heated to approximately 25 °C and mixed using an aquarium pump. Calibration over a relatively large temperature difference improves measurement accuracy. The baths are tightly covered, which prevents evaporation. Both baths include a Pt100 sensor, which are connected to the DTS machine. Temperature is calibrated using the 'DTScalibration' Python library (des Tombe and Schilperoort, 2019) described and used in des Tombe et al. (2020), configured for Weighted Least Squares (WLS) double ended calibration with calibration baths on both ends of the setup. The average measurement uncertainty of temperature data over the whole dataset is equal to 0.13 °C after calibration, according to the Monte Carlo estimate.

#### 3.2.2 Field observations






A section of the resulting calibrated along-cable temperature data over the height of the coil is depicted in Figure 7. Figure 7a is supplemented by a plot of the net radiation at the surface, as observed by the KNMI radiation sensor. The figure shows a change in net radiation as a result of a change in cloud cover, both at night and during the day. These events can be recognized by sudden increases or decreases in net absorbed radiation at the surface. The coil temperatures closely follow these changes over time as a result of the high temporal measurement resolution. Figure 7b illustrates the vertical variability in the temperature profiles that the coil captures as a result of the spatial resolution. In this figure, local temperature minima are found close to the top of the grass (at approximately 0.1 m), which is likely the result of increased radiative cooling in this region. The region just above the surface forms a relatively constant warm layer, illustrating the insulating properties of the canopy/mulch layer. In both figures, faint horizontal bands appear across the time axis, which are caused by the presence of the frame support rings. These artifacts will be further discussed in subsection 3.3.

**Figure 7.** (a) Calibrated temperature data over time, as gathered by the coil. The coil data is supplemented by net absorbed radiation data at the surface, as measured by KNMI at CESAR. As a result of changes in cloud cover, a change in absorbed radiation is observed, both at night and during the day. These events occur roughly at 01:00 and 10:30 respectively. The coil temperature data captures this effect at a high temporal resolution. (b) Data with high variability along the vertical coordinate. Due to the high vertical resolution of the coil, subtle differences along the vertical coordinate can be resolved. The coldest temperatures are found close to the top of the grass layer. A relatively warm layer is found just above the surface, which remains almost constant. Note that the time scale is much shorter here than in panel (a)

For further analysis, temperature profiles are time-averaged over 30 minutes to reduce noise. The data is supplemented by the KNMI air temperature measurement at 1.5 m and 0.1 m for validation. Additionally, more supplementary parameters are included. These are: cumulative precipitation over a 30 minute interval in mm (P), surface net radiation balance in W/m<sup>2</sup> (Q) and total windspeed in m/s (U).

Figure 8a depicts an example of a stably stratified temperature profile at night. Due to the absence of solar radiation, the surface is cooling, causing cold air to stratify above the surface. As described in section 1, the classical description of these temperature profiles is logarithmic. However, alternative, more physically robust models are being developed. The field observations presented here are useful for assessing the performance of these alternative models. The presence of a strong near-surface gradient in the data further illustrates the importance of these high-resolution observations. For qualitative comparison it is interesting to extrapolate the profile and compare it to the KNMI observations at standard height (1.5 m). This may indicate whether there are large biases in absolute temperatures measured by the coil. The profile is extrapolated by fitting the data to a logarithmic relation with free parameters a and b of the following form:  $T = a \cdot ln(z) + b$ . Only the top 50 cm of data is used for extrapolation to mitigate the influence of non-trivial surface effects. In this instance, deviations from the validated KNMI data remain within the DTS measurement error.

Additionally, non-trivial cases are found in the data as well. Figure 8b shows an inversion of the temperature profile within the grass layer. Obviously, the overall profile in Figure 8b is non-trivial and deviates significantly from a purely logarithmic form that is assumed in most numerical parametrizations. Note that the temperature minimum occurs within the grass layer, only several cm above the surface, which would have made it difficult to capture in the past. With the current method, approximately 80 data points are located within the grass canopy. As a result, internal canopy effects can be resolved effectively. The local temperature inversion can possibly be explained by radiative cooling at the top of the grass, in combination with increasing density of grass elements towards the surface (higher up, only few grass elements 'stick out', as can be seen in Figure 5d). As lower sections of the grass are exposed to a lesser degree, radiative cooling will be lower in these sections. As a result, the minimum temperature occurs just below the top of the grass. Note that formally the top of the grass does not exist, because in reality, heights of grass elements are variable (Figure 5d). Similar inversions have been observed before for taller canopies like maize and forests (Jacobs et al., 1992; Jim, 2011). The data demonstrates a strong influence of canopy transport dynamics on near-surface temperature profiles. Therefore, physical understanding of these regions is important for improving near-surface temperature models. Canopy heat transport models can be evaluated or potentially developed using a data-driven approach, on the basis of the observations presented here (Van Der Linden et al., 2022).

#### 3.3 Environmental artifacts and biases

The data validation shows that, in most cases, the coil temperature profiles are consistent with the KNMI point observations which adhere to World Meteorological Organization (WMO) standards. However, there are a number of cases in which the coil temperature deviates from the actual air temperature. Environmental effects such as precipitation and solar radiation cause biases and artifacts on temperature profiles, as expected from similar campaigns (Sigmund et al., 2017). These artifacts and

**Figure 8.** (a) A stable stratified temperature profile during the nighttime. Supplementary KNMI data is included, including precipitation, radiation, wind speed, and air temperature at 1.5 and 0.1 m. The grass canopy region is shaded in light green, which should be considered to be indicative of grass height, rather than absolute. The temperature profile consists of 800 data points in the lowest meter above the surface, 80 of which are located within the canopy. The coil temperature profile is logarithmically extrapolated to 1.5 m to compare it to KNMI air temperature, as to allow a qualitative comparison. The temperature at 0.1 m can directly be compared to the supplementary data. (b) Non-trivial temperature profile with a local temperature inversion at the top of the grass layer. It can be resolved due to the abundance of in-canopy data, revealing the effects of in-canopy transport dynamics.

biases also occur in the field data. As the campaign lasted several months, the most significant contributions to biases can be identified and quantified by comparison with benchmark KNMI data.

Figure 9a shows an example temperature profile after a precipitation event. Because the coil is wet, both the fiber and the frame lose heat by evaporative cooling. As a result, the profile is biased towards colder temperatures by approximately 0.3 °C, with respect to the KNMI air temperature measurements.



Furthermore, colder temperature artifacts appear at the six locations where the support rings are. At these locations, a water residue can build up on the horizontal ring surfaces. This local water storage results in evaporative cooling that is stronger and longer-lasting, with respect to other sections of the coil. Support ring artifacts are also observed at times where the coil is affected by radiative effects. The support rings introduce an increased surface area, causing the coil to locally absorb or emit more radiation. Fortunately, due to the high probing resolution support ring artifacts can be removed (by slicing or other outlier removal methods) from the dataset while retaining a sufficient number of data points to obtain a reliable vertical profile. Depending on circumstances, data for about 3 cm of coil height has to be omitted for the ring artifacts, which translates to 12 data points per ring. In the worst case, this results in 18% data loss.

Figure 9b depicts an example of a case where solar radiation was strong. A significant temperature bias towards warmer temperatures is observed when comparing the profile with KNMI data at 1.5 m height. This bias is a result of both the fiber

optic cable and the frame absorbing radiation. The absolute bias is stronger than the precipitation example and can reach up to 1 °C. Apart from an overall shift towards warmer temperatures, a higher horizontal variability in the data is observed. Radiation causes a horizontal temperature gradient across the coil, that changes with the solar radiation incidence angle. As there are approximately 4 measurements per revolution, temperatures oscillate at the winding frequency in case of a horizontal gradient. Horizontal variability diminishes at the top of the coil, where radiation reaches both sides of the coil equally. As a result of this top artifact, simple purely logarithmic temperature extrapolation becomes less accurate, as can be seen by the fitted dashed line overlaying the data. This further contributes to the observed large temperature bias. Radiative effects on DTS fibers have been widely studied and corresponding data can give insight into the accuracy of observed temperatures (de Jong et al., 2015). Hence, it is recommended that coil observations are accompanied by traditional (shielded) temperature and radiation observations as to estimate the magnitude of potential biases.

**Figure 9.** (a) Temperature profile after rainfall. The full profile is shifted to colder temperatures with respect to the validation measurement. This occurs due to evaporative cooling. Furthermore, the support rings result in cold spike-like artifacts, as they store water and remain cold longer. (b) Temperature profile during a high solar radiation event. The profile is biased toward warmer temperatures and features artifacts at the support ring locations. Radiative effects have a much lower heating impact at the canopy level, due do shading effects. Furthermore, a spread in the data is observed, which is caused by a horizontal temperature gradient on the coil. This gradient results from one side of the coil being irradiated while the other side remains shaded. The spread effect diminishes toward the top of the coil, where both halves absorb radiation due to the angled incidence of solar radiation.

The DTS profile deviation from the KNMI data was evaluated at every time step, both at 0.1 m and 1.5 m height. This analysis shows that the median absolute value of these errors is equal to 0.23 °C and 0.34 °C respectively. For both biases, radiation was found to form the most important contribution to large outliers at 1.5 m height. A clear linear relationship can be identified between net absorbed radiation and temperature bias (Figure 10a), with a Pearson correlation coefficient of  $\rho$ =0.68. This discrepancy is due to the fact that the KNMI reference temperature observations used radiation-shielded temperature probes, which reduces the effects of radiative heating. Radiation effects were quantified using a linear fit. This fit yields a



**Figure 10.** (a) The difference in extrapolated coil temperature and KNMI temperature at 1.5 m height, against net radiation at the surface. A linear relationship exists between temperature error and net radiation. As KNMI air temperature sensors are radiation shielded, the coil records higher temperatures. (b) The difference in coil temperature and KNMI temperature at 0.1 m height, against net radiation at the surface. A weak positive correlation between net radiation and temperature bias appears to be present. However the correlation is significantly weaker than at 1.5 m. Canopy shading effects explain the smaller bias at this height.

radiative bias of  $2.55*10^{-3}$  m<sup>2</sup> K/W. Finally, radiation is less strongly correlated to temperature bias at 0.1 meters, with correlation  $\rho$ =0.23, as is observed in Figure 10b. This can likely be attributed to shading effects of the grass canopy. Fitting the data to a linear expression at this height is not justified, due to the weak correlation.

# 330 4 Design optimization






The functional design was directly taken into the field as it passed all essential threshold criteria in the lab tests. However, there are several design aspects which can be further improved in future campaigns. Some recommendations for future improvements are given below.

In the initial design stage, the decision was made to omit the bottom support ring, as that section of the coil would already be supported by the ground stakes that anchor the coil to the ground. However, the tensile stress on the coil, due to the winding of the fiber, turned out to be much larger than initially expected. Therefore a wooden auxiliary ring was slotted over the ribs to support the coil during the winding process. Removing this ring after installation compromises the structural integrity of the coil as a result of the stress on the frame. It is therefore recommended to always include a permanent bottom support ring on the coil.

Furthermore, some revision on the fiber clamping mechanism is worth investigating. The current design uses luster screw terminals with rubber dampers to fasten the fibers. They provide excellent grip on the fiber, however, the screws impact the integrity of the cable. Signal loss was detected on one of the two luster terminals, as a result of over-tightening. Signal losses can be compensated to a certain degree by using a double ended configuration on the DTS. Nonetheless, it is advantageous to keep signal loss to a minimum to keep the signal to noise ratio as high as possible. To this end, it is recommended that the Lester terminals are fitted with more resilient dampers and that they are not over-tightened. Other means of fastening the fiber can also be investigated, keeping in mind the need for relatively high clamping force to keep the fiber at the required tension. Winding the coil inflicts a torque on the coil. With each revolution, the fiber pulls the frame in the direction of winding. This can cause the frame to experience torsional deformation over the full length of the coil. It is important to be mindful of this when winding the coil, as these effects can be mitigated by properly supporting the ribs while winding. Torsion within the frame does not impact the measurements. However, it may impact the structural integrity of the frame.

Finally, removing the coil from the field was a challenge, due to the fact that deeply protruding soil stakes were glued to the frame after installation. As all stakes were put into the soil separately, They all had slightly different orientations, making removal near impossible. The stakes were glued to the frame for redundancy. However, merely placing the frame on top of the stakes gives it sufficient rigidity, given that the top of the coil is tethered to the ground. It is recommended to keep the frame-stake connections glue free, as this makes removal much easier. This also allows the coil to be reused for multiple campaigns. Alternatively, one could opt for shallow stakes which are already attached to the frame in the file generation process. In combination with anchor lines, this will also give the coil sufficient stability, while retaining the possibility of removing and reusing it.

As mentioned in the introduction, FRADTS was developed with the FAIR guiding principles in mind (Wilkinson et al., 2016). These principles are aimed at open data sharing, but can also be applied to software (Hut et al., 2022). In this paper, we aim to further extend these principles to a hardware application. The guiding principles are: Findability, Accessibility, Interoperability and Reusability. An important novel approach for this design was its reusability. Seeing as it is unlikely for researchers to want to identically reproduce this exact setup, the script can be reused and adapted to a wide range of other

applications, beyond the implementation described here. Therefore, we believe that the design aligns well with this guiding principle. Furthermore, FRADTS has been made as accessible as possible, openly providing the full script with the freedom to adapt and reproduce the laser files, though we acknowledge that Rhinoceros is a paid software package.

# 5 Conclusions





The FRADTS method proved to be successful for probing temperature profiles at millimeter resolution. The method supports the design and construction of coil frames with minimal effort and manufacturing experience. The design is parametric, so physical characteristics of the desired setup can be easily modified, allowing application of FRADTS in many different scenarios. Furthermore, a setup is uniquely defined by its input parameters. Reporting which parameters were used for a setup yields sufficient information to recreate the identical setup. To validate the performance of the proposed method, a coil frame was manufactured and deployed in the field for several months. A vertical resolution and accuracy of 1.25 mm was achieved. The temperature profiles are representative of local air temperature, with median absolute errors of 0.23 °C and 0.34 °C at 0.1 m and 1.5 m height respectively. However, external factors such as precipitation and radiation absorption and emission have a noticeable effect on the data. The most significant contribution is that of solar radiation absorption, which can introduce biases of over 1 °C. Furthermore, support ribs introduce spike-like artifacts in the data under the aforementioned conditions. Artifacts can, to a certain degree, be selectively removed from the dataset. Biases can be estimated using supplementary local weather data, in particular radiation and precipitation. We note that such weather-related artifacts are not unique to this setup, as radiation biases affect all outdoor DTS-measurement setups.

Despite environmental effects, FRADTS is proven to be effective in surveying spatial temperature profiles at high resolutions and closely corresponds to air temperature in most cases. The method has multiple use cases. For example, FRADTS can be deployed above a grass canopy to quantitatively evaluate and compare RSL models (Boekee et al., 2024) or to develop incanopy heat transport models (Van Der Linden et al., 2022), as was done in the CESAR field campaign. Moreover, FRADTS is applicable in other regions as well, potentially proving useful in profiling temperatures in other short canopies and above lakes (Paaijmans et al., 2008), snow (Zeller et al., 2021), ice (De Bruijn et al., 2014), animal burrows (Ganot et al., 2012) or urban areas (Hong et al., 2024).

Code availability. A repository containing instructions and the laser file generation code is hosted in a github repository: https://github.com/cgbterHorst/DTS-based-coil-measurement-technique

The specific version of the repository used for generation of the coil as presented in this paper is published on Zenodo (ter Horst, 2025)

Author contributions. Contributions are reported in accordance with NISO CRediT contributor roles: C.G.B.t.H. developed and executed the design (software) and corresponding experimental methodology, formally analyzed resulting data and validated it, wrote the original

manuscript draft, implemented edits on the basis of review, and was responsible for submission. Part of this manuscript is based on the master thesis by the first author (ter Horst, 2024). G.A.V. conceptualized the research project and experimental methodology, had the leadership responsibility on experimental work and reviewed the manuscript. J.J-B. had an overall advisory/supervision role and reviewed the manuscript. R.W.H. had an advisory/supervision role in the design process and reviewed the manuscript. B.J.H.v.d.W. was the primary supervisor and instructor, worked on the overall project conceptualization and reviewed and edited the manuscript.

Competing interests. The authors declare that they have no conflict of interest.


Acknowledgements. The authors acknowledge M. Koning, G. van Wegen and R. Ronda for their help in setting up the field campaign at Cabauw and for allowing us to make use of the facilities of the KNMI. The authors acknowledge S. van Dijk for helping with setting up the field campaign. Furthermore, the authors acknowledge T. Haanstra, D. la Haye and R. Breider from the Makerspace at the department of Applied Physics for offering their facilities and knowledge during the course of this project.

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
