# Peer review of "An Adaptable DTS-based Parametric Method to Probe Near-surface Vertical Temperature Profiles at Millimeter Resolution"

_EGUsphere, 2025_

## Referee Comment (RC1)

**Review - egusphere-2025-1397**

June 2025

**1 General Remakes**

The manuscript introduces a reproducible method of developing a coiled-DTS array capable of observing air temperature at the millimeter scale. The authors go through various constraints necessary for a good design and present a material capable of fulfilling these design requirements. A parametric method for developing the DTS coil is developed, from which one can estimate the vertical resolution the array is capable of. The design's vertical accuracy was then verified in lab experiments followed by a field deployment for assessing temperature accuracy and the effect of artifacts. Some exemplary profiles of air temperature were presented. Radiative artifacts were evaluated against a standard reference probe. Extensive documentation for the code and assembly of the DTS coil are provided.

Generally, I think this is a fantastic concept and the manuscript is worthwhile of publication. However, there are a number of items to address first. The writing can be unfortunately repetitive and a general edit is necessary to create a more fluid text. I noted some of these instances. The introduction could do with a bit of reorganizing so that a non-DTS expert can more easily understand the justification of the problem. There are some issues with the description of the lab experiments (e.g., potentially flipped axes, saying experiments will be discussed later and not discussing them). The biggest need is creating a more robust statistical comparison against the reference probes, otherwise the statements being made are too ambitious given the limited results shown. The documentation is extensive and commendable. One thing to potentially add is a piece of code that converts the DTS from LAF to height. I look forward to seeing the revisions as I believe this work is important and provides a powerful method.

**2 Major Comments**

**Introduction**:

- The paragraph starting on line 57 introduces concepts that a non-DTS expert would need earlier to understand the discussion of previous work.

I recommend moving this concept to be much earlier, especially since this paragraph in essence introduces the entire problem and makes the literature review clearer.

- I also recommend expanding the studies that used coiled DTS setups in your review of previous work to better incorporate studies not originating from the same institution (e.g., more directly include Sigmund et al and Zeller et al).

- Line 82-84: I was left confused because the cited studies did study the specified media but the sentence suggests these media have not been studied.

**Design**

- Line 134: Citing a thesis, while accepted, should be done only when strictly necessary. I am certain you did great work in it, but I do not want to read a thesis to understand your manuscript. If the design considerations were not relevant enough to include in the manuscript then I suggest not mentioning them at all. If they are important, they should be discussed, even if briefly, in the text. I do not want to diminish the work you did, but I also do not want to read another document.

- Material selection: It seems like there is a trade off between the materials depicted in gold and blue in Figure 3. The materials in blue minimize thermal conductivity while those in gold have the potential to further minimize heat capacity at the expense of larger thermal conductivity. Could you comment on the reason to minimize one over the other, e.g., how was the "pareto front" chosen?

- The parametric design lists coil radius as one of the critical parameters. But, this parameter is also limited by the minimum bend allowed by the fiber. I think including a short warning of that limitation would be beneficial.

**Experimenal validation and discussion**

- Naively, based on the affiliation of the authors of the study, I would assume you are using the 'dtscalibraton' python package. If so, please include a citation for the paper describing the method as well as a citation for the code. Upon further reading I see that I am correct in which case the information needs to be consolidated.

- The lab experiment for assessing the vertical accuracy has the water level decreasing with time in sub panel (c) while the water level appears to be increasing in time in sub-panel (b). Could you please clarify what the axes mean? Additionally, how was the true water level assessed?

- End of section 3.1: A lab experiment for assessing the effect of rain was performed and a later analysis of the experiment was promised in Section 3.3, but no such analysis was presented.

- Section 3.2.1: For describing the field setup it is also necessary to include the separation distance between the observations. How consistent is the grass height between the reference temperature probe and the DTS coil? Please address.

- DTS uncertainty versus resolution: On line 109 the instrument time and temperature resolutions are specified, but it is known this is different than the instrument accuracy. Later this is recognized through the calibration bath validation, with an instrument uncertainty of 0.13 C. I think the introduction would benefit from highlighting the literature assessing the actual uncertainties and resolvable scales in addition to the manufacturer supplied resolutions.

- 3.2.2 and Figure 5: I am left puzzling if the DTS device is measuring artifacts from the grass contacting the DTS. Could you comment on this either here or in the manuscript?

- 3.2.2: Time-averaging to 30 minutes is a substantial amount of time aggregation. In night time conditions the temperature structures will almost certainty include processes at minute time scales, as seen in Figure 7b. Further, this is a total of 180 observations (correct?), which seems like an unnecessary number of observations needed in order to "reduce noise". I recommend carefully evaluating if this level of temporal aggregation is necessary.

- 3.2.2: I am quite certain I read much of this material in the introduction.

- Figure 8 and 9: Given the logarithmic shape and the fine scale features, I recommend moving these plots to an $ln(z)$ spacing. Further, the variability of the observations should be indicated (I anticipate the variability will be large which is part of the reason I think a 30 minute average is inappropriate).

- 3.2.2: I strongly disagree that extrapolating an observation to a distance 100% outside the fitted region counts as a validation, as suggested in the Figure caption and in this section. If you want to make this statement, I strongly argue for the inclusion of a more robust statistical fit, including uncertainty as well as a statistical test comparing the extrapolated value to the reference observation. In fact, I think I would like to see a statistical comparison generally. The 'statsmodel' in python is a useful package for performing such statistical inferences and tests. I also think a general illustration of the error distribution is necessary.

- The paragraph starting on line 311 needs to be re-written. It is currently too informal for a publication. Further, many of the assessments come off as overly confident given the sparsity of information presented.

- Section 3.3: There is no mention of wind-mitigated radiation artifacts. Including an analysis on the basis of wind and net radiation simultaneously

would benefit the statements being made regarding the accuracy of the system. It also seems like the Sigmund et al., 2017 manuscript could also be cited in this section. Finally, it is stated that radiative effects can be compensated for, but this was not performed here, which seems a bit odd.

**Conclusions**

- Line 405-406: It is stated that this resolution and accuracy have never been achieved before, but this was not discussed directly in the text and seems hard to verify in any case. Specifically, part of the motivation for the study was that it is hard to verify what the resolution and accuracy was for other studies. I recommend amending this statement to be consistent with the motivation.

**3 Minor Comments**

- The abstract reads a bit disjointed and could benefit from making the sentences flow better into each other.

- Line 10: "different, identical" I think this sentence needs to be clarified.

- Line 34: Zeller et al do use a coiled DTS setup, but they do not specify this is to observe the insulating plant canopies as implied.

- Line 52-53: Vertical accuracy is unclear here. I think re-organizing the introduction as suggested could help make the intent clear.

- Line 118: I do not believe that you need to make the parenthetical statement.

- Line 164-166: I naively would assume that reducing the specific heat would reduce the lag between a temperature change and the change in the signal observed by DTS.

- Line 181: The sentence starting here needs revision.

- Section 3.2.1: Many sentences begin with "This site" or similar. Please re-write to be less repetitive.

- Line 264-266: The sentences are disjointed and incomplete.

- Line 319: "the" is an odd choice for a word to bold.

- Line 341: "In the worst case" and "sub-optimal conditions" convey the same concept.

- Line 417: I am confident I read this statement previously in the manuscript.

---

## Referee Comment (RC2)

Review of:

**An Adaptable DTS-based Parametric Method to Probe Near-surface Vertical Temperature Profiles at Millimeter Resolution**

Constantijn G.B. ter Horst, Gijs A. Vis, Judith Jongen-Boekee, Marie-Claire ten Veldhuis, Rolf W. Hut, and Bas J.H. van de Wiel

**Journal:** *Atmospheric Measurement Techniques*

ter Horst and coauthors describe the design strategy, their own implementation, and experimental validation of a novel technique for distributed temperature sensing (DTS) at unprecedented resolution with a tightly wound fiber-optic cable. They take an open-source approach, with design instructions and laser file generation freely available, which I find especially admirable for a project like this which could feasiblly be kept proprietary and marketed as a private instrument for sale. Overall, I think the instrument is useful and well designed, and in my review I hope to elevate the reach that it could have to the many communities that surely need finer resolution temperature measurements. After minor revisions I can support the publication of this work.

I will start my review by saying that I am not a regular reader of *Atmospheric Measurement Techniques* which I think is relevant based on some of the feedback that I have about writing style and target audience. Specifically, my two major points of feedback are:

1) The article is too fixated on a single use case for the instrument. I am aware that this use case for measuring near surface air temperature in a short canopy is particularly relevant for the readership of the journal, but I can imagine the instrument being useful in so many different ways, and you do mention a few of them (e.g., L82-84). To name a few that would be relevant in my work in the cryosphere:
    a. soil temperature
    b. snow temperature (gradients particularly important for avalanche forecasting)
    c. sea-surface temperature gradient (set up this platform on a floating bouy)
    d. sea-floor temperature gradient
    e. ice-shelf front (temperature gradients and micro-currents are extremely important for understanding melt and currently poorly understood).

The point here is not that you need to fully describe every possible use case, but your introductory content should not limit the scope of possible uses. Lead with a paragraph that is generally about how there are often very strong temperature gradients at natural material boundaries but those are generally poorly measured (which is true), then describe DTS in the way you have and reserve and finally that you chose one relevant case as field validation which happened to be this grass canopy environment.

2) The manuscript *feels* like an instruction manual rather than a scientific article. That may not be a terrible thing, especially for this journal (as I said I am not a regular reader), but it may be a bit dry for a lot of readers. If you choose to limit the instruction manual feel, my suggestions would be to:
   a) Cut down the design section in favor of the reader's attention on subsequent sections which I do think feel more scientific and narrative driven.
   b) Turn the lists for threshold and optimization criteria into a table or schematic.
   c) Remove some of the overly specific details such as this sentence: "When the desired cutout path is achieved, the file is exported in an .SVG file." I don't think that the file type is particularly important in a narrative style article but would be important in the instruction manual. I give some other similar examples in the line items below.

I also think it would be neat if you had a fun name for this device, and could consider including that as a part of your title. You call it "the coil" or refer to "the design" and "the frame" throughout, but it would capture the attention of more readers if you had a strong name for the design/devise. Something like:

Fine Resolution Adaptable Distributed Temperature Sensing (FRADTS)

Ha, I don't know, just an idea.

**Line items**

Title - "parametric" is sort of jargon-y in the sense that you are using it and pretty much means the same thing as adaptable.

L8 – "down to" instead of up to?

L8-9 – This sentence about the laser cutout path needs something to make it clear that it is for creating the instrument frame. Something like: "Our method uses a parametric script to specify the laser cutout path for the instrument frame components are assembled in a coil-like structure to hold the DTS fiber."

L10 – "different" and "identically reproducible" are close to each other in an awkward way. I would say that the parameters can be changed to "customize the design" and always in a reproducible way.

L16 – Successful based on what? Specifics here will be more convincing to a reader.

Overall, I would say the abstract should focus more on your FAIR approach and how that could make the product useful to many communities, and don't lead with thermal properties of the grass environment, save that for your description of the field test case.

L106-107 – The numbers 5 mm and 2 mm seem arbitrary here. Is there a physical reason you chose those (i.e., based on the environment to measure)? Or this is just a reasonable goal and you believe your design wouldn't be a sufficiently significant improvement from other methods if this prescribed resolution were not met.

L120-121 – That the temperature measurements should be consistent with the temperature of the medium feels like it should be a threshold criteria to me, perhaps because of how it is phrased? If you dropped this sentence (or moved it to threshold) then the rest of this bullet makes it more clear that you are want to 1) minimize thermal mass to lower the equilibration time of the intrsument and frame to the temperature of the medium, and 2) minimize the conductivity so that the frame is not moving heat across the temperature gradient you are trying to measure.

L131-132 – Can you give estimates for cost and fabrication time here? "minimized" is vague.

L135-139 – The description of laser cutting is wordy and not really needed here. Just a brief statement that you have open source laser cutting files and maybe a reference to the laser cutting technique if that exists, those would suffice.

L170 – Just call it the "threshold criteria" since you set that up above, no reason to change to new language for "discrete constraints".

L181 – "we" not capitalized

L181 – "generation" and "generates" feel weird together, and what is a "new generation script" anyway, just say your approach is novel.

Equation 1 – I didn't fully appreciate until getting to here that you are treating this as a 1-dimensional measurement. That is, that the temperature variation within one coil wrap is effectively averaged over because the along-cable resolution is more like 25 cm, as you say in the abstract. It may be worth more plainly stating this and the assumptions that go with it (i.e., that you are looking for scenarios with a strong temperature gradient in only a single direction).

L192 – "[of] a given step"?

Table 1 caption – Restate "cable height" with the 1000 mm to make it clear that is what you are talking about (third sentence).
1.25 mm resolution along the cable? Is that true or am I misunderstanding what you are stating here? In the abstract you say 25 cm.

L196 – I don't think that this sentence adds anything of substance. Describe the fabrication and installation and the user can decide for themselves whether it is simple or if they will need some patience.

L204 – How important is it that the winding is consistent or at a specific tension? Does the cable need to be precisely lined up with the wrap above and below it?

L208 – I would save this reference of Figure 5, and perhaps even the mention of the field test, for section 3.2. It feels weird that Figure 5 comes before Figure 6.

L249 – CESAR acronym never defined

L262 – Interrogator malfunction? Or?

L264 – the statement that "the data is still considered sufficient for validation" would be stronger if it was explicitly linked to the next statement: "sufficient for validation because…"

L272 – Is that described in Figure 5? Maybe include some annotation to make it more clear what you are talking about here.

L315 – Is this parenthetical exclamation intentional? I am not sure it is appropriate for this writing style.

L319 – is the bold intentional?

L327 – WMO acronym not defined.

L332-334 – The measurements are also significantly more variable than in the non-rain case. Do you have a simple explanation for that?

L339 – These can be removed, agree, but presumably you would agree that the vertical support structure has the same effect as the horizontal rings and that is much more difficult to remove (also possibly more problematic as it moves heat in the vertical as you mention).

L346 – I am confused about this discussion on horizontal variability since I see more variability in 9a than 9b and that was not mentioned.

L353 – but at 1.5 meters it is an extrapolation instead of a true measurement, correct? Need to say that if true.

**Figures**

Figure 2. You say that fiber position and spacing is accurately defined "as can be seen in the image", but it is not entirely clear what you mean by that. You are saying that the spacing between fiber wraps is consistent? Maybe some annotations on the image here would be helpful.

Figure 6. It is not entirely clear how you are extracting the location of maximum gradient from the temperature data. You calculate a numerical gradient between points and select the maximum? And how are the uncertainty bars which you plot calculated? Adding a 1-d temperature plot that indicates the maximum gradient might be helpful here.

Figure 7. More notes and annotations added to the figure would be helpful. For instance, it is not immediately obvious to a reader what is night/day so adding annotations for those would help them see that instantly, also that the horizontal scale between (a) and (b) is very different. Add an arrow pointing out the very thin insulated layer in the grass at the bottom of (b).

Figure 8. Am I understanding correctly that the large blue dot at the top of the grass is a measurement but that at 1.5 m is the extrapolation? If so, I would suggest plotting them differently. Perhaps consider plotting the full line that you are extrapolating, from your measurements to 1.5 m continuously.

Figure 9a. Is the anomalous gradient at the top of your profile one of the "spike-like" artifacts caused by the rings? That one is particularly prominent and warrants more description.

---

## Author Comment (AC2)

*ter Horst and coauthors describe the design strategy, their own implementation, and experimental validation of a novel technique for distributed temperature sensing (DTS) at unprecedented resolution with a tightly wound fiber-optic cable. They take an opensource approach, with design instructions and laser file generation freely available, which I find especially admirable for a project like this which could feasibly be kept proprietary and marketed as a private instrument for sale. Overall, I think the instrument is useful and well designed, and in my review I hope to elevate the reach that it could have to the many communities that surely need finer resolution temperature measurements. After minor revisions I can support the publication of this work.*

*I will start my review by saying that I am not a regular reader of \*Atmospheric Measurement Techniques\* which I think is relevant based on some of the feedback that I have about writing style and target audience.*

We thank you for your peer review on our manuscript. We appreciate the encouraging words on the applicability of the method we have developed. We have addressed your suggestions and discussed them below. If a comment has been highlighted in green, this means that it has been directly implemented in the manuscript without the need for further comment.

*Specifically, my two major points of feedback are:*
*1) The article is too fixated on a single use case for the instrument. I am aware that this use case for measuring near surface air temperature in a short canopy is particularly relevant for the readership of the journal, but I can imagine the instrument being useful in so many different ways, and you do mention a few of them (e.g., L82-84). To name a few that would be relevant in my work in the cryosphere:*
*a. soil temperature*
*b. snow temperature (gradients particularly important for avalanche forecasting)*
*c. sea-surface temperature gradient (set up this platform on a floating bouy)*
*d. sea-floor temperature gradient*
*e. ice-shelf front (temperature gradients and micro-currents are extremely important for understanding melt and currently poorly understood).*
*The point here is not that you need to fully describe every possible use case, but your introductory content should not limit the scope of possible uses. Lead with a paragraph that is generally about how there are often very strong temperature gradients at natural material boundaries but those are generally poorly measured (which is true), then describe DTS in the way you have and reserve and finally that you chose one relevant case as field validation which happened to be this grass canopy environment.*

Indeed we have mentioned some other applications in other regions (e.g., L82-84 and L417-L420). However, we explicitly choose to keep the current narrative on account of the journal in which we aim to publish and our own expertise on the interpretation of these observations

*2) The manuscript feels like an instruction manual rather than a scientific article. That may not be a terrible thing, especially for this journal (as I said I am not a regular reader), but it may be a bit dry for a lot of readers. If you choose to limit the instruction manual feel, my suggestions would be to:*
*a) Cut down the design section in favor of the reader's attention on subsequent*

*sections which I do think feel more scientific and narrative driven.*
*b) Turn the lists for threshold and optimization criteria into a table or schematic.*
*c) Remove some of the overly specific details such as this sentence: "When the*
*desired cutout path is achieved, the file is exported in an .SVG file." I don't*
*think that the file type is particularly important in a narrative style article but*
*would be important in the instruction manual. I give some other similar*
*examples in the line items below.*

We agree with the fact that some parts of the design section dive into too much detail, which should indeed be left to the instruction manual. Some overly specific lines in this section have been deleted or simplified.

Also, we agree with your suggestion to reformat the criteria into a table. They are now presented in a more organized way.

*I also think it would be neat if you had a fun name for this device, and could consider including that as a part of your title. You call it "the coil" or refer to "the design" and "the frame" throughout, but it would capture the attention of more readers if you had a strong name for the design/devise. Something like:*
*Fine Resolution Adaptable Distributed Temperature Sensing (FRADTS)*
*Ha, I don't know, just an idea.*

Indeed, we have thought about a suitable acronym for a long time, but did not come up with anything both catchy and meaningful. It is difficult to find an acronym that encompasses so many different terms 🙂. However, we really liked your suggestion for and decided to include it in the manuscript. The method is now referred to as FRADTS and most instances of 'the method' have been replaced by 'FRADTS' (not all to avoid being repetitive in some sections. Note that this is the name for the method as a whole (as that is what we are presenting) and that we still refer to 'the coil' where just the physical frame is concerned. Finally, as some key words are missing in 'FRADS' that are important to the method, we chose to not use the acronym for the title of the manuscript, but rather introduce it later in the text. It is however mentioned in the abstract.

*Title - "parametric" is sort of jargon-y in the sense that you are using it and pretty much means the same thing as adaptable.*

We do agree that these two terms convey similar concepts. However, we prefer to keep the term 'parametric' as it stresses the fact that this design is centered around a parametric design script. Many different physical parameters can be quickly adjusted. We believe that "parametric" conveys this concept a bit better than "adaptable".

*L8 – "down to" instead of up to?*

*L8-9 – This sentence about the laser cutout path needs something to make it clear that it is for creating the instrument frame. Something like: "Our method uses a parametric script to specify the laser cutout path for the instrument frame components are assembled in a coil-like structure to hold the DTS fiber."*

*L10 – "different" and "identically reproducible" are close to each other in an awkward way. I would say that the parameters can be changed to "customize the design" and always in a reproducible way.*

*L16 – Successful based on what? Specifics here will be more convincing to a reader.*

*Overall, I would say the abstract should focus more on your FAIR approach and how that could make the product useful to many communities, and don't lead with thermal properties of the grass environment, save that for your description of the field test case.*
We have decided to fully rewrite the abstract, taking into account this comment and that of other reviewers.

*L106-107 – The numbers 5 mm and 2 mm seem arbitrary here. Is there a physical reason you chose those (i.e., based on the environment to measure)? Or this is just a reasonable goal and you believe your design wouldn't be a sufficiently significant improvement from other methods if this prescribed resolution were not met.*
Indeed, the 5 mm and 2 mm are somewhat arbitrary, as they are not derived from anything physical. As stated in your comment: we believe that we can speak of a significant improvement over older methods if these requirements are met.

We did add a line:
"... accuracy required for the application at hand. For the application presented here, steep temperature gradients in a grass layer, we set the minimum requirements at 5 and 2 mm, respectively."

*L120-121 – That the temperature measurements should be consistent with the temperature of the medium feels like it should be a threshold criteria to me, perhaps because of how it is phrased? If you dropped this sentence (or moved it to threshold) then the rest of this bullet makes it more clear that you are want to 1) minimize thermal mass to lower the equilibration time of the intrsument and frame to the temperature of the medium, and 2) minimize the conductivity so that the frame is not moving heat across the temperature gradient you are trying to measure.*
The suggestion is very well phrased and explains the concepts well. We therefore chose to add it to the criterion. We do however believe that this criterion is best categorized as an optimization. For example, figure 3 clearly illustrates that an optimization is required to attain the best material for the design.

*L131-132 – Can you give estimates for cost and fabrication time here? "minimized" is vague.*
When setting up criteria, we aim to be specific and brief. Giving examples or estimates within a criterion will move it towards the threshold category, as a certain cost or fabrication time may not be exceeded. Rather, we prefer to say that we try to keep these aspects as low as possible in the design.
We understand that 'minimized' may be associated with a mathematical optimization, which is not done here. However, we feel this is the best way to phrase optimization criteria

*L135-139 – The description of laser cutting is wordy and not really needed here. Just a brief statement that you have open source laser cutting files and maybe a reference to the laser cutting technique if that exists, those would suffice.*
We believe this section to be important in developing an understanding of the methods used. Especially since this is the central technique around which this method revolves, we prefer to keep this section, in spite of the fact that it may be slightly wordy. Finally, many sentences

such as "Laser cutters are available at most modern workshops and makerspaces" supply direct proof of the fact that the accessibility criterion has been met, which is important to explicitly state.

*L170 – Just call it the "threshold criteria" since you set that up above, no reason to change to new language for "discrete constraints".*

*L181 – "we" not capitalized*

*L181 – "generation" and "generates" feel weird together, and what is a "new generation script" anyway, just say your approach is novel.*

*Equation 1 – I didn't fully appreciate until getting to here that you are treating this as a 1-dimensional measurement. That is, that the temperature variation within one coil wrap is effectively averaged over because the along-cable resolution is more like 25 cm, as you say in the abstract. It may be worth more plainly stating this and the assumptions that go with it (i.e., that you are looking for scenarios with a strong temperature gradient in only a single direction).*

We've specifically added "one-dimensional" to the first criterion to further stress the fact that we are only measuring temperature along the vertical.

*L192 – "[of] a given step"?*

*Table 1 caption – Restate "cable height" with the 1000 mm to make it clear that is what you are talking about (third sentence).*

*1.25 mm resolution along the cable? Is that true or am I misunderstanding what you are stating here? In the abstract you say 25 cm.*

Indeed you are right that this is a typo. Thanks for catching it. It has been corrected.

*L196 – I don't think that this sentence adds anything of substance. Describe the fabrication and installation and the user can decide for themselves whether it is simple or if they will need some patience.*

We have slightly shortened this sentence to take away some unnecessary text. However, we disagree that this sentence is not important. We aim to stress that any researcher can build such a setup without the need for great technical skills. We believe this point to be of great importance, as otherwise readers may be dissuaded from trying this method in their own groups.

*L204 – How important is it that the winding is consistent or at a specific tension? Does the cable need to be precisely lined up with the wrap above and below it?*

While these aspects are generally important, the design makes sure of all these things already. This is due to the notches on the ribs which ensure proper alignment. Furthermore, tension merely needs to be sufficient for the fiber to stay on the coil, which is very obvious while winding the coil. We therefore chose not to explicitly state these things as they are not important to anyone wanting to create a coil.

*L208 – I would save this reference of Figure 5, and perhaps even the mention of the*

*field test, for section 3.2. It feels weird that Figure 5 comes before Figure 6.*

The aim behind figure 5, and the setup description section as a whole, is to describe all details pertaining to the physical construction and setup of the coil, before doing any measurement. Figure 6 illustrates the configuration hooked up for measurement, which should logically come only after the physical installation in our opinion.

*L249 – CESAR acronym never defined*

*L262 – Interrogator malfunction? Or?*

Unfortunately, the exact cause of the malfunction is still not known to us. During the surveying period the DTS machine would occasionally shut down. As with many things, a software update seems to have fixed it :)

*L264 – the statement that "the data is still considered sufficient for validation" would be stronger if it was explicitly linked to the next statement: "sufficient for validation because…"*

*L272 – Is that described in Figure 5? Maybe include some annotation to make it more clear what you are talking about here.*

"Described" was not the correct word to use here. We have changed it to 'depicted' to more closely match the intended meaning.

*L315 – Is this parenthetical exclamation intentional? I am not sure it is appropriate for this writing style.*

We were not sure to include it, even amongst ourselves. It was intended to stress the remarkably large amount of data within the canopy. However, we realise it is more appropriate to leave out the exclamation point and have removed it from the manuscript.

*L319 – is the bold intentional?*

Same as the comment above

*L327 – WMO acronym not defined*

*L332-334 – The measurements are also significantly more variable than in the non-rain case. Do you have a simple explanation for that?*

We were quite shocked to find that, even after multiple rounds of proofreading, the panels in figure 9 were swapped. We highly suspect that this is the reason for this, and other comments. Our apologies for this error.

To specifically address this comment: variability in the radiation case is very high as a horizontal temperature gradient exists on the coil. This is an effect that is not present with rain, as those effects are not horizontally dependent.

*L339 – These can be removed, agree, but presumably you would agree that the vertical support structure has the same effect as the horizontal rings and that is much more difficult to remove (also possibly more problematic as it moves heat in the vertical as you mention).*

Indeed overall biases, as created by the vertical support ribs, are much more difficult to filter out than the artifacts. We believe we addressed this issue with the following line:

"Hence, it
is recommended that coil observations are accompanied by traditional (shielded)
temperature and radiation observations as to
estimate the magnitude of potential biases."

*L346 – I am confused about this discussion on horizontal variability since I see more variability in 9a than 9b and that was not mentioned.*
Again, this is a result of the swapping of figures.
Once more, our apologies for the confusion.

*L353 – but at 1.5 meters it is an extrapolation instead of a true measurement, correct? Need to say that if true.*
Indeed we will work to make this more obvious. We will redo the plots and change from a blue point to a dotted line for the extrapolations.

*Figure 2. You say that fiber position and spacing is accurately defined "as can be seen in the image", but it is not entirely clear what you mean by that. You are saying that the spacing between fiber wraps is consistent? Maybe some annotations on the image here would be helpful.*
We've added to the figure caption:

"The distance between the fibers is very consistent, meaning that its position is precisely defined."

*Figure 6. It is not entirely clear how you are extracting the location of maximum gradient from the temperature data. You calculate a numerical gradient between points and select the maximum? And how are the uncertainty bars which you plot calculated? Adding a 1-d temperature plot that indicates the maximum gradient might be helpful here.*
The following was added:

"The position of this large gradient can be found by calculating the numerical gradient and tracking the spatial position of its maximum."

"The mean absolute error is taken to be the positional accuracy of the configuration."

To avoid cluttering the already crowded figure, we decided against including another panel. Rather, we prefer further clarification in text, as was done by the aforementioned lines.

*Figure 7. More notes and annotations added to the figure would be helpful. For instance, it is not immediately obvious to a reader what is night/day so adding annotations for those would help them see that instantly, also that the horizontal scale between (a) and (b) is very different. Add an arrow pointing out the very thin insulated layer in the grass at the bottom of (b).*

We would argue that the time scale under the plot is sufficient for the reader to assess when night/day occurs and to get a sense of temporal scale. We chose to add a small note in the caption. Furthermore, we believe the bright yellow indication of an insulating layer is sufficient to draw the readers attention to this region. While we understand your comment, we prefer not to clutter the figure.

*Figure 8. Am I understanding correctly that the large blue dot at the top of the grass is a measurement but that at 1.5 m is the extrapolation? If so, I would suggest plotting them differently. Perhaps consider plotting the full line that you are extrapolating, from your measurements to 1.5 m continuously.*

As mentioned before, the plots will be changed to a dotted line for improved clarity. As you mention, this communicates more clearly that we are extrapolating data.

*Figure 9a. Is the anomalous gradient at the top of your profile one of the "spike-like" artifacts caused by the rings? That one is particularly prominent and warrants more description.*

This effect arises as a result of radiative effects. We see the spread in temperatures (which are a result of a horizontal temperature gradient due to half the coil being shaded by itself) decrease towards the top of the coil, as the sun hits both sides of the coil here. The height at which this happens depends on the solar zenith angle. At the top, the temperature profile dramatically dips to a cooler temperature at the top, as a result of the top support ring. This effect is explained in the manuscript as follows:

"Furthermore, a spread in the data is observed, which is caused by a horizontal temperature gradient on the coil. This gradient results from one side of the coil being irradiated while the other side remains shaded. The spread effect diminishes toward the top of the coil, where both halves absorb radiation due to the angled incidence of solar radiation."

---

## Author Comment (AC3)

**RC1 rebuttal**

**An Adaptable DTS-based Parametric Method to Probe Near-surface Vertical Temperature Profiles at Millimeter Resolution**

*The manuscript introduces a reproducible method of developing a coiled-DTS array capable of observing air temperature at the millimeter scale. The authors go through various constraints necessary for a good design and present a material capable of fulfilling these design requirements. A parametric method for developing the DTS coil is developed, from which one can estimate the vertical resolution the array is capable of. The design's vertical accuracy was then verified in lab experiments followed by a field deployment for assessing temperature accuracy and the effect of artifacts. Some exemplary profiles of air temperature were presented. Radiative artifacts were evaluated against a standard reference probe. Extensive documentation for the code and assembly of the DTS coil are provided.*
*Generally, I think this is a fantastic concept and the manuscript is worthwhile of publication. However, there are a number of items to address first. The writing can be unfortunately repetitive and a general edit is necessary to create a more fluid text. I noted some of these instances. The introduction could do with a bit of reorganizing so that a non-DTS expert can more easily understand the justification of the problem. There are some issues with the description of the lab experiments (e.g., potentially flipped axes, saying experiments will be discussed later and not discussing them). The biggest need is creating a more robust statistical comparison against the reference probes, otherwise the statements being made are too ambitious given the limited results shown. The documentation is extensive and commendable. One thing to potentially add is a piece of code that converts the DTS from LAF to height. I look forward to seeing the revisions as I believe this work is important and provides a powerful method.*

We thank you for your peer review on our manuscript. We appreciate the encouraging words on the merit of the method we have developed. Your comments have been extensively discussed and are individually addressed below. If a comment has been highlighted in green, this means that it has been directly implemented in the manuscript without the need for further comment.

*2 Major Comments*
*The paragraph starting on line 57 introduces concepts that a non-DTS expert would need earlier to understand the discussion of previous work. I recommend moving this concept to be much earlier, especially since this paragraph in essence introduces the entire problem and makes the literature review clearer.*

We agree with the reviewer that this paragraph needs revision. After some internal discussion we decided to move the introduction of the coil geometry to an earlier section, while giving a short description of the geometry:

"Therefore, adapted, compactly positioned fiber optic configurations were implemented to attain sufficiently high resolution vertical temperature profiles. Fiber optic cable was attached to a net to form a two-dimensional harp-like structure. Additionally, novel configurations have been developed where helically wound coils further increase vertical fiber density (Hilgersom, 2016, Sigmund, 2017, Izett, 2019, Zeller, 2021). These campaigns implemented rigid, vertical frames, around which the fiber optic cable is wound. As a result of these enhanced geometries, resolutions and accuracies up to the centimeter scale were achieved."

Furthermore, we agree that the explanation of the geometrical resolution limitations was convoluted. We changed this paragraph to the following:

The vertical probing resolution is a critical limitation in quantifying temperature gradients. Previous campaigns have achieved resolutions of several centimeters, but capturing near-surface gradients within the upper meter requires sub-centimeter resolution and accuracy. Beyond the inherent spatial resolution of DTS devices, this limitation is largely geometric in nature and can be divided into two aspects. The first is that of **vertical fiber density**: increasing the length of fiber per unit vertical distance enhances resolution. This issue is addressed by the enhanced fiber geometries, such as the aforementioned coil geometry. The second aspect is that of **positional accuracy**: the temperature measurement at any point depends directly on the precise location of the fiber, which requires the cable to be fixed securely in space. Even with increased vertical density, uncertainties in fiber positioning remain a limiting factor. Thus, further increasing fiber winding density in isolation is insufficient; both vertical density and positional accuracy must simultaneously be addressed to achieve meaningful improvements in vertical resolution.

*I also recommend expanding the studies that used coiled DTS setups in your review of previous work to better incorporate studies not originating from the same institution (e.g., more directly include Sigmund et al and Zeller et al).*
These studies are now more explicitly mentioned and explained

*Line 82-84: I was left confused because the cited studies did study the specified media but the sentence suggests these media have not been studied.*
We have rephrased this section to more explicitly state that these media have been studied, but not at such high resolution:
"This potentially extends such high resolution observations to applications beyond the grass region towards other media, such as snow (Zeller, 2021), ice (deBruijn, 2014) or water surfaces (Paaijmans, 2008)."

*Line 134: Citing a thesis, while accepted, should be done only when strictly necessary. I am certain you did great work in it, but I do not want to read a thesis to understand your manuscript. If the design considerations were not relevant enough to include in the manuscript then I suggest not mentioning them at all. If they are important, they should be discussed, even if briefly, in the text. I do not want to diminish the work you did, but I also do not want to read another document.*

The reviewer is right: the present manuscript already contains the full information for the reader to repeat the experiments. Also reference to a master thesis should be minimized as those typically are less accessible. However, we supply the thesis as additional reading material for interested readers. We suggest the following:

We removed the old line, which might suggest that the thesis contains essential material for the understanding of the method, not already described in the manuscript, which is false.

As some readers might be interested to see more photos of the laboratory setup we added a reference in the caption of figure 4: "note that more photo material on the manufacturing and lab-tests is available in ter Horst (2024)."

Likewise we added a sentence in the authors contribution:
"part of this manuscript is based on the master thesis by the first author (ter Horst, 2024). ..."
This is sufficient for the reader interested in further details, while it is not essential indeed for the general reader.

*Material selection: It seems like there is a trade off between the materials depicted in gold and blue in Figure 3. The materials in blue minimize*
*thermal conductivity while those in gold have the potential to further minimize heat capacity at the expense of larger thermal conductivity. Could*
*you comment on the reason to minimize one over the other, e.g., how was*
*the "pareto front" chosen?*

Defining a meaningful pareto front, often referred to as the 'material index' in ashby plots, is difficult, if at all possible. Relating the two parameters in a mathematical way, such that a slope can be found is not trivial and requires extensive physical modeling. Luckily, in Figure 3 it is observed that all of our relevant non-grey materials align under each other vertically. As such, picking blue material is still defendable for practical reasons here. PMMA comes with many additional practical advantages, apart from its thermal properties. These are described in text and as such constitute a sufficient justification of the material choice, in our opinion.

*The parametric design lists coil radius as one of the critical parameters.*
*But, this parameter is also limited by the minimum bend allowed by the*
*fiber. I think including a short warning of that limitation would be beneficial.*
*Experimental validation and discussion*

We added:
"Note that the minimum coil radius is constrained by the minimum bend radius allowed by the type of fiber that is used."

*Naively, based on the affiliation of the authors of the study, I would assume*
*you are using the 'dtscalibraton' python package. If so, please include a*
*citation for the paper describing the method as well as a citation for the*
*code. Upon further reading I see that I am correct in which case the*
*information needs to be consolidated.*

Here we refer to line 276 where we listed the paper by Des Tombe. However, we will add an additional citation to the code, in addition to the paper that is already cited. Furthermore, we specified that it is a python library.

*The lab experiment for assessing the vertical accuracy has the water level*

*decreasing with time in sub panel (c) while the water level appears to be increasing in time in sub-panel (b). Could you please clarify what the axes mean? Additionally, how was the true water level assessed?*

The z-coordinate is relative to the coil itself, where z=0 is at the bottom of the coil (the side with the stakes). The coil was positioned upside down in the container, as this was more convenient. We will add a note clarifying this in the figure caption.

The water level is assessed by measuring the start and end level, under the assumption that the rise in water level is linear.

*End of section 3.1: A lab experiment for assessing the effect of rain was performed and a later analysis of the experiment was promised in Section 3.3, but no such analysis was presented.*

The reviewer is right that no description is given on the results of the lab tests, as the observed artifacts also appear in field data. Describing these observations twice would be superfluous. The initial motivation for mentioning the lab tests is that they isolated environmental factors, making it easier to link them to observed artifacts. This allowed us to more confidently explain our observations in the field. However, we acknowledge that this is not of significant interest to the reader. We have therefore removed all mentions of the environmental lab tests and describe artifacts only in the field observation section.

*Section 3.2.1: For describing the field setup it is also necessary to include the separation distance between the observations. How consistent is the grass height between the reference temperature probe and the DTS coil? Please address.*

Indeed the reviewer is right here. The grass-height is maintained and mowed at an average height of approximately 10 cm. However, grass is a living material and the 10cm is only approximate.
While we inspected both sites which looked very similar, there is no absolute 'proof' (as for Wimbledon-grass or soccer grass). In the text we will therefore add a disclaimer sentence:

line 288 (at approximately ~ 0.1 m height).

And line 251" The site is well suited for observations as it is located in a relatively homogeneous, mostly agricultural area. with grass at the Cabauw site that is mowed and maintained at approximately ~ 0.1m height (though it is noted here that grass height may vary somewhat and as such the 0.1m is more indicative than absolute).

And in the figure caption: "The grass canopy region is shaded in light green, which should only be considered indicative, rather than absolute (section 3.21.).

*DTS uncertainty versus resolution: On line 109 the instrument time and temperature resolutions are specified, but it is known this is different than the instrument accuracy. Later this is recognized through the calibration bath validation, with an instrument uncertainty of 0.13 C. I think the introduction would benefit from highlighting the literature assessing the actual uncertainties and resolvable scales in addition to the manufacturer supplied resolutions.*

Although we see the point by the reviewer, we disagree here, that we should address more literature on this, as for the goal of the paper, the present quantification sufficiently supports the conclusions. Adding a lot of extra detail and discussion here would distract from the main message.

Note that apart from an order of magnitude (mm rather than cm as previously) it is not justified to give very hard numbers, as multiple (unknown) sources of uncertainty will play an additional role.

*3.2.2 and Figure 5: I am left puzzling if the DTS device is measuring*
*artifacts from the grass contacting the DTS. Could you comment on this*
*either here or in the manuscript?*
This is a valid comment. However, it is virtually impossible to quantify. Fortunately we observe no large outliers in the grass layer and spikes such as with the wetted rings. As such we suspect no major role. We will however keep this in mind for our follow up research.

*3.2.2: Time-averaging to 30 minutes is a substantial amount of time aggregation. In night*
*time conditions the temperature structures will almost*
*certainty include processes at minute time scales, as seen in Figure 7b.*
*Further, this is a total of 180 observations (correct?), which seems like*
*an unnecessary number of observations needed in order to "reduce noise".*
*I recommend carefully evaluating if this level of temporal aggregation is*
*necessary.*
In this case, we choose our data aggregation time on the basis of the KNMI data, which we use for validation. As such, we have to aggregate to half-hour intervals. Though, higher time resolutions are definitely possible with this setup and such large aggregations are not necessary. It would be interesting to look at shorter averaging times and 'times-to convergence' as to study e.g. nighttime intermittency in turbulence, in a style similar to the study of hartogensis et al, fig. 6, who did that for scintillometry.

Boundary-Layer Meteorology 105: 149–176, 2002.

*3.2.2: I am quite certain I read much of this material in the introduction.*
Indeed, we have shifted some essential information to the introduction and made a reference to the introduction in section 3.2.2.

*Figure 8 and 9: Given the logarithmic shape and the fine scale features,*
*I recommend moving these plots to an ln(z) spacing. Further, the variability of the*
*observations should be indicated (I anticipate the variability*
*will be large which is part of the reason I think a 30 minute average is*
*inappropriate).*
Indeed, we explicitly considered this in our research group (plotting ln(z)), however, we decided to plot the 'unmodified/raw' data, as this is the most direct way of plotting. Plotting ln(z) plots can also 'hide' uncertainties. Moreover, the main purpose of this paper is to present technical methods rather than in-depth scientific analysis. A more in-depth scientific

analysis will be the subject of a separate paper that is currently under development in our group.

*3.2.2: I strongly disagree that extrapolating an observation to a distance 100% outside the fitted region counts as a validation, as suggested in the Figure caption and in this section. If you want to make this statement, I strongly argue for the inclusion of a more robust statistical fit, including uncertainty as well as a statistical test comparing the extrapolated value to the reference observation. In fact, I think I would like to see a statistical comparison generally. The 'statsmodel' in python is a useful package for performing such statistical inferences and tests. I also think a general illustration of the error distribution is necessary.*

We agree with the reviewer. The confusion may be caused by our use of the word 'validation', which is too strict in a scientific sense. The extrapolation is much more a qualitative comparison, as it uses a theory/model (i.e. the log-law) in itself. Indeed, the comparison is outside the raw observational range.

We therefore change our text. We delete:
"indicating the validity of the measurement."

We add: "The coil temperature profile is
logarithmically extrapolated to 1.5 m to compare it to KNMI air temperature, as to allow a qualitative comparison (see text)."

Also in the text we modify :
"The validity of the measured profile is underscored by the air temperature measurements done by the KNMI (grey dot), as compared to the coil-extrapolated profile (blue dot)".
To:
"For qualitative comparison it is interesting to extrapolate the profile and compare it to the KNMI observations at standard height (1.5m). This may indicate whether there are large biases in absolute temperatures measured by the coil."

Finally, we changed the plots by substituting the extrapolated point by a dashed line.

As we intend for qualitative comparison only here to inspect large absolute biases a full statistical analysis (apart from those already in the paper Figs 10 on this. ) is beyond the scope of the present work.

*The paragraph starting on line 311 needs to be re-written. It is currently too informal for a publication. Further, many of the assessments come off as overly confident given the sparsity of information presented.*

The sense of informality and perhaps overconfidence is perhaps triggered the the exclamation mark behind the number 80 (line 315) and the 'bold' printing in line 319. In fact this was point of discussion between the authors themselves. So those 2 items will be removed.
We do think that the core of the paper is very innovative in a sense that such in-grass inversions and in-grass temperature dynamics have never been reported before, so that

some excitement in the language is not out of place. But yet, it should be scientific language of course, on that we agree.

We also modify: "Such inversions have qualitatively been observed before at much lower resolutions (Jacobs et al., 1992; Jim, 2011)." which may sound overconfident by:

"Similar inversions have been observed before for taller canopies like maize and forests (e.g. Jacobs et al., 1992; Jim, 2011)."

*Section 3.3: There is no mention of wind-mitigated radiation artifacts.*
*Including an analysis on the basis of wind and net radiation simultaneously would benefit the statements being made regarding the accuracy of the*
*system. It also seems like the Sigmund et al., 2017 manuscript could also*
*be cited in this section. Finally, it is stated that radiative effects can be*
*compensated for, but this was not performed here, which seems a bit odd.*
We have investigated the effect of wind on the temperature bias, but did not find a significant correlation that was worth presenting.
An analysis linking temperature bias to both radiation and wind simultaneously would be interesting in future campaigns, but remains outside the scope of this research for now.

Indeed this paragraph is a good place to add a reference to Sigmund et al., 2017. It was added in this paragraph.

We have also addressed the line about compensation and changed it to:

"Radiative effects on DTS fibers have been widely studied and corresponding data can give insight into the accuracy of observed temperatures (deJong, 2015). "

Which more accurately represents the role of radiation data in our research.

*Line 405-406: It is stated that this resolution and accuracy have never*
*been achieved before, but this was not discussed directly in the text and*
*seems hard to verify in any case. Specifically, part of the motivation for the*
*study was that it is hard to verify what the resolution and accuracy was*
*for other studies. I recommend amending this statement to be consistent*
*with the motivation.*
We deleted the part: "which has not been attained before using DTS-based setups.'
Indeed the resolution in the order or 1-2 mm may already show the potential enough. No need to emphasize more than that (in accordance with your earlier comment line 311 paragraph.).

*The abstract reads a bit disjointed and could benefit from making the*
*sentences flow better into each other.*
We have decided to fully rewrite the abstract, taking into account this comment and that of other reviewers.

*Line 10: "different, identical" I think this sentence needs to be clarified.*

*Line 34: Zeller et al do use a coiled DTS setup, but they do not specify this is to observe the insulating plant canopies as implied.*

*Line 52-53: Vertical accuracy is unclear here. I think re-organizing the introduction as suggested could help make the intent clear.*

*Line 118: I do not believe that you need to make the parenthetical statement.*

*Line 164-166: I naively would assume that reducing the specific heat would reduce the lag between a temperature change and the change in the signal observed by DTS.*

Indeed, this is the case. We added a small note of this for further clarity:

"Reducing the amount of heat stored within the frame diminishes the potential heat transfer to the fiber optic cable, decreasing equilibration time and thus improving response speed."

*Line 181: The sentence starting here needs revision.*

*Section 3.2.1: Many sentences begin with "This site" or similar. Please re-write to be less repetitive.*

*Line 264-266: The sentences are disjointed and incomplete.*

*Line 319: "the" is an odd choice for a word to bold.*

*Line 341: "In the worst case" and "sub-optimal conditions" convey the same concept.*

*Line 417: I am confident I read this statement previously in the manuscript.*